# Economic motivation for raising coastal flood defenses in Europe

Michalis I. Vousdoukas [1✉], Lorenzo Mentaschi [1], Jochen Hinkel [2,3], Philip J. Ward [4], Ignazio Mongelli[1], Juan-Carlos Ciscar[1] & Luc Feyen[1]

Extreme sea levels (ESLs) in Europe could rise by as much as one metre or more by the end of this century due to climate change. This poses significant challenges to safeguard coastal communities. Here we present a comprehensive analysis of economically efficient protection scenarios along Europe's coastlines during the present century. We employ a probabilistic framework that integrates dynamic simulations of all ESL components and flood inundation, impact modelling and a cost-benefit analysis of raising dykes. We find that at least 83% of flood damages in Europe could be avoided by elevating dykes in an economically efficient way along 23.7%-32.1% of Europe's coastline, specifically where high value conurbations exist. The European mean benefit to cost ratio of the investments varies from 8.3 to 14.9 while at country level this ranges between 1.6 and 34.3, with higher efficiencies for a scenario with high-end greenhouse gas emissions and strong socio-economic growth.

---

[1] European Commission, Joint Research Centre (JRC), Ispra, Italy. [2] Global Climate Forum, Adaptation and Social Learning, Neue Promenade 6, Berlin 10178, Germany. [3] Division of Resource Economics at Albrecht Daniel Thaer-Institute and Berlin Workshop in Institutional Analysis of Social-Ecological Systems (WINS), Humboldt-University, Berlin, Germany. [4] Institute for Environmental Studies (IVM), Vrije Universiteit Amsterdam, 1081 HV Amsterdam, the Netherlands. ✉email: Michail.VOUSDOUKAS@ec.europa.eu

The coastal zone is an area of high interest. At present, more than 200 million European citizens live within 50 km from the coastline, stretching from the North–East Atlantic and the Baltic to the Mediterranean and Black Sea, and current trends indicate that migration toward coastal zones is continuing[1,2]. Coastal areas host important commercial activities and also support diverse ecosystems that provide important habitats and sources of food[3]. Coastal zones are particularly vulnerable to climate change due to the combined effects of sea level rise and potential changes in the frequency and intensity of storms[4,5]. Global mean sea level has increased by 13–20 cm since pre-industrial times[6]. This process has accelerated since the 1990s[9,10] due to global warming[7]. This has already contributed to coastal recession[8,9] and made Europe's coasts more susceptible to coastal hazards. The continued rise in sea levels along Europe's coastlines in view of global warming could result in unprecedented coastal flood losses in Europe, in case no additional coastal protection and risk-reduction measures are implemented[10].

There exists a range of possible adaptation measures to increase the resilience of future coastal societies to flooding[11], summarized as protect, accommodate, retreat and do nothing[12]. The latter option seems unrealistic considering the substantial presence of critical infrastructure along the global coastline[13,14], and the projected economic losses from coastal flooding[10,15]. Retreat involves relocating dwellings and infrastructure in order to reduce coastal flood risk[16], but relocation is often challenging due to public opposition or practical limitations (e.g. moving critical infrastructure such as ports of power plants)[1,17]. Accommodate involves reducing the damages from coastal flooding, either through effective forecasting/warning systems and emergency response[18], or flood proofing of structures[12]. Protect involves preventing flooding through natural (dunes) or artificial (dykes) structures, beach nourishment, and nature-based solutions, which have recently gained attention as more environmentally sustainable ways to protect and maintain coastlines[19].

Among the various adaptation options, hard protection is the strategy that delivers the most predictable levels of safety against coastal extremes and sea level rise and has been applied widely along developed coastlines of Europe[20], despite the fact that hard protection can affect the landscape in a negative way, increase erosion, reduce amenity value and result in more catastrophic events in the case of failure[21]. As a result, there is the increasing trend to apply a combination of adaptation options, with some kind of impermeable protection element being one of them.

Here, we evaluate the costs and benefits of applying additional protection through dyke improvements along the European coastline, assuming that the densely populated and high income European coastal communities will choose to hold the line. We employ a probabilistic data and modelling framework, that includes the following steps: first, estimate present and future extreme sea levels along Europe's coastlines based on state-of-the-art projections of sea level rise, waves, storm surges and tides for a high emissions (RCP8.5) and moderate emissions (RCP4.5)[22] pathway; second, delineate the land areas inundated when extreme sea levels overtop current coastal protection and derive the corresponding flood inundation depth using 2-D hydraulic modelling[14]; third, overlay the inundation maps with exposure information on population and land use; fourth, translate this into direct flood losses using functions that relate the depth of inundation with economic damage to the assets inundated, and into the number of people flooded, taking into account gridded socio-economic projections[1], according to the shared socio-economic pathways (SSPs; expressing changes in asset values and level of urbanization in the future)[23]; fifth, following the same approach to assess European river flood risk[24], repeat steps two to four with step-wise increases in dyke height and compute

economic benefits (= avoided flood damage during this century) and dyke cost based on unit costs taken from the literature; finally, for each coastal segment the dyke height that maximizes the net present value (NPV), which is the discounted sum of the dyke cost (negative) and the economic benefits (positive) over the project lifetime, is considered the optimum. Dyke costs include construction investment and maintenance costs. Benefits are the avoided losses from coastal flooding from 2020 up to the end of this century. We applied a discount rate of 5% for the EU cohesion countries and 3% for the other EU member states (see Methods).

One of the important advances of the present study is that it resolves the spatial dependencies of extreme sea levels (ESLs) along the European coastline, considering temporally dynamic contributions not only from SLR, but also astronomical and meteorological components, throughout the century. All variables computed in this study are available as probability density functions, but we focus our discussion mainly on the expected values, as well as on the very likely range, represented by the 5th–95th quantiles. The analysis considers the uncertainty range of damages and costs. Results are presented and discussed at four spatial scales: along ~10,000 coastal sections of the European coastline, as well as at NUTS2, country and European level. We assess two scenarios: (i) sustainability, combining moderate emissions with SSP1, which represents global sustainable development; and (ii) fossil fuel development, combing high emissions with SSP5[23]. The highest increase in GDP and population is projected under fossil fuel development, reaching by the end of the century values which are 10 and 2 times higher than the baseline, respectively. Population in 2100 is similar to the baseline under sustainability, while GDP is almost triple.

Our study underlines the unprecedented rise in damages from coastal flooding unless mitigation and adaptation measures are taken. We find that around 95% of these impacts could be avoided through moderate greenhouse gas emission mitigation and by raising dykes where human settlements and economically important areas exist along the coastline. The extent to which adaptation can lessen the effects of coastal flooding and at what cost is sensitive to the investment strategy adopted.

## Results

**Sea level rise projections.** The European average 100-year *ESL* is projected to show a very likely increase of 34–76 cm under a moderate-emission-mitigation-policy scenario and of 58–172 cm under a high emissions scenario. The above increase is similar to the one projected for the mean sea level at European level. The biggest *ESL* increase is projected for the North Sea, due to increasing meteorological contributions, while the contrary applies for the northern Baltic Sea; due to land uplift. The Atlantic coast of Spain and Portugal is also projected to experience a lower increase in extreme sea levels, due to milder storms[25].

**The consequences of doing nothing.** At present, coastal flood losses in Europe amount to €1.4 billion per year (all values are expressed in 2015 € values), and each year about 100,000 EU citizens are affected from coastal flooding. Flood risk is projected to increase strongly in Europe with global warming. In the absence of further investments in coastal adaptation and under the Sustainability scenario, annual coastal flood losses for Europe by the end of the century are projected to grow to €209.8 billion around a very likely range from €29.7 billion to €844.5 billion (Table 1). Similar estimates under fossil fuel based development are €1268.4 billion (160.9–4731.1). During the same time span, the total number of people flooded in Europe is projected to rise

**Table 1 Direct damages without any future adaptation measures.**

| | Baseline | Sustainability | Fossil fuel based development |
|---|---|---|---|
| Belgium (BE) | 0.0 | 4.5 | 20.7 |
| Bulgaria (BG) | 0.0 | 0.1 | 0.6 |
| Cyprus (CY) | 0.0 | 1.4 | 12.5 |
| Germany (DE) | 0.1 | 6.0 | 38.8 |
| Denmark (DK) | 0.0 | 8.9 | 84.6 |
| Estonia (EE) | 0.0 | 0.1 | 0.6 |
| Spain (ES) | 0.1 | 9.4 | 53.0 |
| Finland (FI) | 0.0 | 0.3 | 6.2 |
| France (FR) | 0.2 | 40.4 | 266.0 |
| Greece (GR) | 0.1 | 4.9 | 20.8 |
| Croatia (HR) | 0.0 | 0.9 | 2.5 |
| Ireland (IE) | 0.1 | 14.5 | 89.1 |
| Italy (IT) | 0.1 | 15.3 | 70.3 |
| Lithuania (LT) | 0.0 | 0.2 | 0.4 |
| Latvia (LV) | 0.0 | 0.1 | 0.2 |
| Malta (MT) | 0.0 | 0.0 | 0.0 |
| The Netherlands (NL) | 0.0 | 20.7 | 77.5 |
| Norway (NO) | 0.1 | 25.6 | 200.1 |
| Poland (PL) | 0.1 | 2.3 | 9.3 |
| Portugal (PT) | 0.1 | 2.2 | 8.7 |
| Romania (RO) | 0.0 | 0.1 | 2.8 |
| Sweden (SE) | 0.0 | 4.2 | 46.2 |
| Slovenia (SI) | 0.0 | 0.7 | 2.9 |
| United Kingdom (UK) | 0.4 | 47.2 | 254.3 |
| Europe | 1.4 | 209.8 | 1268.4 |

Expected annual damage (EAD, in billion €) from coastal flooding in 2100 under sustainability with RCP4.5 and fossil fuel based development with RCP8.5, shown per country and for Europe. Baseline values are also shown in the first column.

**Table 2 People flooded without any future adaptation measures.**

| | Baseline | Sustainability | Fossil fuel based development |
|---|---|---|---|
| Belgium (BE) | 0.5 | 13.1 | 31.7 |
| Bulgaria (BU) | 0.6 | 1.6 | 2.4 |
| Cyprus (CY) | 3.0 | 13.8 | 17.2 |
| Germany (DE) | 2.0 | 33.6 | 113.3 |
| Denmark (DK) | 1.0 | 79.2 | 273.4 |
| Estonia (EE) | 0.1 | 0.3 | 0.8 |
| Spain (ES) | 8.1 | 182.4 | 346.1 |
| Finland (FI) | 0.5 | 3.5 | 39.8 |
| France (FR) | 3.5 | 145.9 | 393.2 |
| Greece (GR) | 10.7 | 81.0 | 144.2 |
| Croatia (HR) | 9.2 | 31.3 | 41.6 |
| Iran (IR) | 3.1 | 104.4 | 237.6 |
| Italy (IT) | 12.7 | 200.9 | 382.3 |
| Lithuania (LT) | 1.3 | 4.2 | 6.3 |
| Latvia (LV) | 0.2 | 0.4 | 0.8 |
| Malta (MT) | 0.0 | 0.1 | 0.2 |
| The Netherlands (NL) | 1.6 | 5.4 | 12.9 |
| Norway (NO) | 0.1 | 169.6 | 513.8 |
| Poland (PL) | 9.9 | 24.2 | 46.9 |
| Portugal (PO) | 2.6 | 14.4 | 29.7 |
| Romania (RO) | 0.5 | 2.3 | 3.4 |
| Sweden (SE) | 0.5 | 33.4 | 124.5 |
| Slovenia (SI) | 2.4 | 6.2 | 10.0 |
| United Kingdom (UK) | 27.7 | 461.3 | 1126.1 |
| Europe | 100 | 1612.6 | 3898.2 |

Expected annual number of people flooded (EAPF, in thousand people) by coastal flooding in 2100 under sustainability with RCP4.5 and fossil fuel based development with RCP8.5, shown per country and for Europe. Baseline values are also shown in the first column.

to 1.61 (0.5–3.1), and 3.9 million (3.9–6.9), for the two scenarios, respectively (Table 2). Coastal flood risk will increase in all EU-countries that have a coastline, with France, the UK, Italy and Denmark showing the highest absolute increase in coastal flood risks towards the end of the century. For some countries, coastal flood losses at the end of this century could amount to a considerable share of their GDP, especially under a high-emissions pathway (RCP8.5), most notable in Cyprus (4.9%), Greece (3.2%), Denmark (2.5%), Ireland (1.8%) and Croatia (1.8%).

**Costs and benefits of adaptation**. While those numbers illustrate the large adaptation needs Europe is facing, the underlying assumption of no further investment in coastal protection is not very plausible given such high relative losses. Conversely, where human life may be at risk and high density, high value conurbations exist, it is very likely that the use of hard defence structures will continue. The benefit to cost ratio (BCR) of increased protection, however, varies strongly across Europe. Costs outweigh benefits for 76% of the European coastline, under Sustainability (likely range 76%–89%), and for 68% under fossil fuel based development (67%–81%; Table 3). This implies that there is no economic motivation for further protecting these areas. Low BCR can be related to several factors, like steep morphology and sparsely populated coastlines, such as in Greece and Malta. Also, long and complex coastlines imply higher dyke construction costs, hence lower BCR, such as in many parts of Finland, Sweden, Estonia, and Croatia. Most of the Baltic is experiencing uplift and, therefore, relative sea level rise is lower compared to other parts of Europe, implying lower increases in future losses and hence potential benefits of adaptation for a significant part of Finland and Sweden.

Despite the absence of economic motivation to adapt along a high percentage of the European coastline, the concentration of human development renders adaptation very economically beneficial in certain areas. Benefits tend to outweigh costs in areas where population density is larger than 500 people per km². In urbanized and economically important areas the benefits tend to exceed the costs by at least an order of magnitude. As a result, when benefits and costs are aggregated at regional level, the total benefits are dominated by those in urban centres and this compensates for the low BCR in less densely populated and rural coastal stretches. At NUTS2 level, adaptation appears as economically efficient (BCR > 1) in about 82% and 86% of the regions, under Sustainability and Fossil Fuel Based Development, respectively. Adaptation comes with far stronger economic benefits in Devon (mean BCR equal to 14 and 60, respectively), Puglia (17 and 49), Murcia (15 and 37), Loire (8 and 44), East Anglia (9 and 44), Languedoc-Roussillon (10 and 42), Merseyside (15 and 31), and Basque country (13 and 33) (Fig. 1).

At country level, Belgium is the country with the highest percentage of coastline where benefits exceed costs (85%–95% depending on the scenario), followed by France (58%–66%), and Italy (53%–59%; Table 3). These are also the countries with some of the highest expected BCRs, varying within 16.6–25.8, 10.5–24.8, and 9.7–16.4, respectively (range expresses variation among scenarios; Fig. 2). Other countries with high BCR values are the Netherlands (Expected BCR between 21.1 and 34.3), Cyprus (11.1–15.6), Ireland (8.8 and 18.7), and Greece (9–11) (Fig. 2 and Table 3). On the lower end of the analysis is Malta, for which the expected country level BCR is the lowest in Europe: 1.6–1.7, depending on the scenario. Other countries with low BCR values are Bulgaria (expected BCR equal to 2–2.1), Lithuania (2–2.1), Latvia (1.8–2.1) and Croatia (1.9–2.3). Since fossil fuel

**Table 3 Benefits vs Costs at country level.**

|  | % Coastline BCR > 1 | | Mean country level BCR | |
|---|---|---|---|---|
|  | Sustainability | Fossil fuel based development | Sustainability | Fossil fuel based development |
| Belgium (BE) | 85.0 | 95.0 | 16.6 | 25.8 |
| Bulgaria (BG) | 5.4 | 8.9 | 2.0 | 2.1 |
| Cyprus (CY) | 22.9 | 27.5 | 11.1 | 15.6 |
| Germany (DE) | 20.9 | 39.1 | 3.4 | 5.8 |
| Denmark (DK) | 22.8 | 48.3 | 3.0 | 6.9 |
| Estonia (EE) | 0.5 | 1.5 | 2.1 | 2.5 |
| Spain (ES) | 46.9 | 56.2 | 8.1 | 15.1 |
| Finland (FI) | 2.1 | 15.5 | 1.7 | 3.3 |
| France (FR) | 58.3 | 66.3 | 10.5 | 24.8 |
| Greece (GR) | 10.7 | 13.0 | 9.0 | 10.5 |
| Croatia (HR) | 8.3 | 10.4 | 1.9 | 2.3 |
| Ireland (IE) | 19.0 | 28.6 | 8.8 | 18.7 |
| Italy (IT) | 52.6 | 59.1 | 9.7 | 16.4 |
| Lithuania (LT) | 4.9 | 9.8 | 2.1 | 2.1 |
| Latvia (LV) | 3.2 | 3.2 | 2.1 | 2.1 |
| Malta (MT) | 6.7 | 13.3 | 1.6 | 1.7 |
| The Netherlands (NL) | 40.1 | 40.8 | 21.1 | 34.3 |
| Norway (NO) | 14.5 | 23.2 | 6.3 | 13.8 |
| Poland (PL) | 24.6 | 30.7 | 3.9 | 4.5 |
| Portugal (PT) | 32.7 | 43.5 | 6.7 | 9.1 |
| Romania (RO) | 3.3 | 14.8 | 2.4 | 2.4 |
| Sweden (SE) | 11.7 | 23.4 | 5.4 | 10.9 |
| Slovenia (SI) | 50.0 | 50.0 | 3.7 | 5.9 |
| United Kingdom (UK) | 25.6 | 33.5 | 7.7 | 14.6 |
| Total | 23.8 | 32.1 | 8.3 | 14.9 |

Percentage of the country's coastline with mean BCR > 1 (benefits of adaptation exceed the costs) and mean country level benefit to cost ratio (BCR) over coastal stretches where additional protection is required. All data are shown for the two scenarios studied: sustainability with RCP4.5 and fossil fuel based development with RCP8.5.

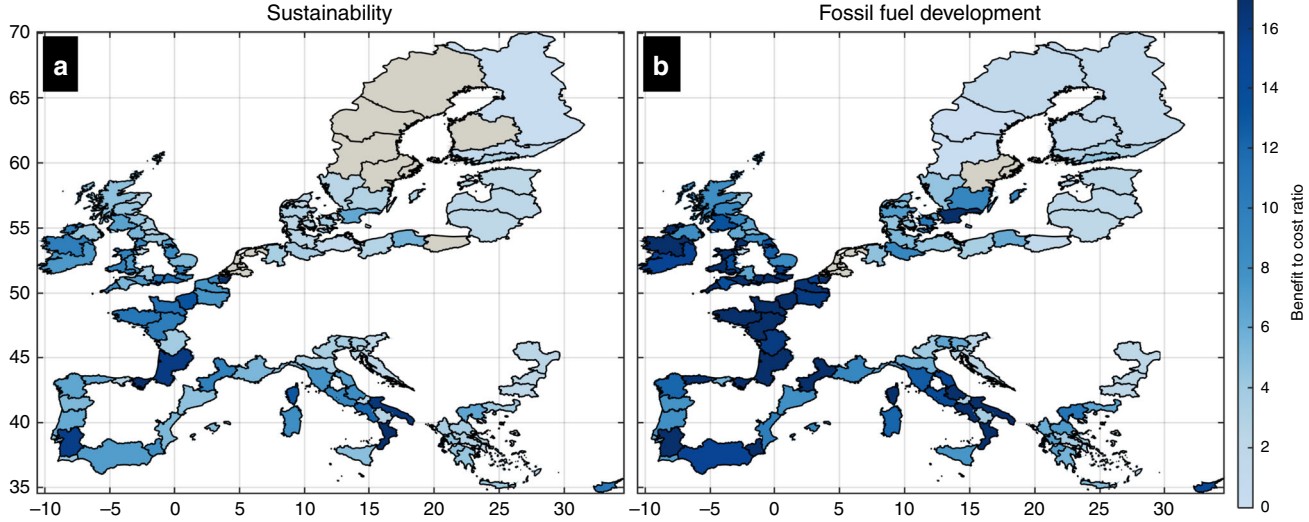

**Fig. 1 Benefits vs Costs at regional level.** Expected benefit to cost ratios of raising coastal protection per NUTS2 region. Values are shown for two scenarios: Sustainability with RCP4.5 (**a**) and fossil fuel based development with RCP8.5 (**b**). Grey colours express areas where the maximum net present value is achieved with the current protection.

based development combines strong increase in ESLs with socio-economic growth, the resulting BCRs are higher, for some countries more than double compared to the other scenario (e.g. France, Ireland, Sweden, Denmark, and Finland (Fig. 2 and Table 3). The mean expected BCR for Europe is 8.3 (likely range: 6.1–17.5) and 14.9 (12.3–29.6), under sustainability and fossil fuel based development, respectively (Fig. 2).

The Netherlands is a particular case as the country is already very well protected (up to ~10,000 year return period), by an

extensive network of dykes and surge barriers. We find that with additional protection it is even less likely that the country will experience a catastrophic flood during the century. However, the Netherlands have a high income level, an extensive low-lying area and high population density, so flood events can have massive impacts when they occur, and for that reason the mean country-level BCR is the highest in Europe. Thus, the benefits from protecting further are high (high mean BCR), even though flood events are rare; explaining the high uncertainty in the BCRs

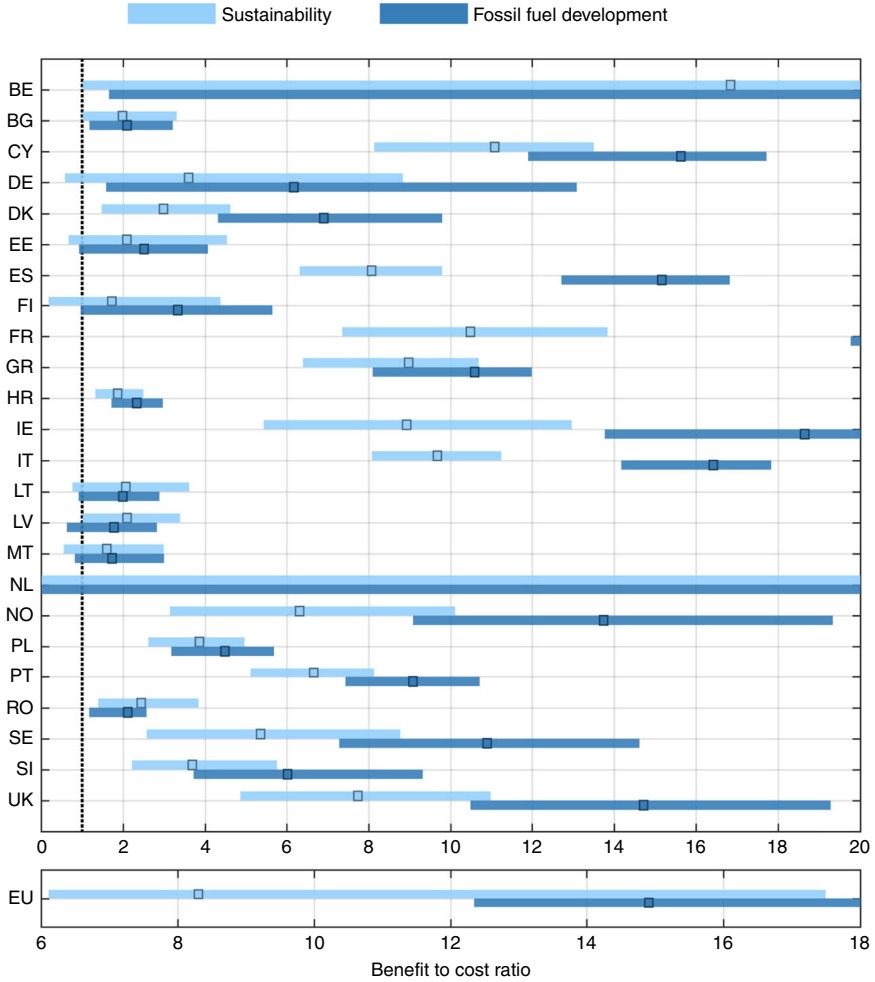

**Fig. 2 Benefits vs Costs at country level.** Each colour expresses a scenario (Sustainability–RCP4.5 (light blue) and fossil fuel based development–RCP8.5 (dark blue)), with patches expressing the very likely range and squares the expected BCR per country. The vertical black dotted line expresses BCR = 1.

(see Fig. 2, while the above sentence applies to a certain extent also for Belgium): among the 10,000 extreme sea level scenarios simulated in the stochastic approach, few included catastrophic floods during the century, resulting in high losses. These rare but high impact events lead to the high expected BCRs. On the contrary, many realizations of future extreme sea levels do not surpass the present high protection standards, hence in such cases flood risk is very low, or even zero in the case of the Netherlands, and so is the low-end BCR.

The expected annual investment on further dyke improvements during the present century, under Sustainability and without discounting, is €1.75 billion per year, around a very likely range of €1.75–€7.39 (Fig. 3 and Table 4). Under fossil fuel based development similar estimates reach €2.82 (very likely range of €2.82–€11.89) billion per year. Country level adaptation costs are mainly controlled by the value of assets and the coastline length, with the UK (€522–719 million per year), France (€269–385 million per year), Norway (€126–296 million per year), Italy (€180–261 million per year), and Germany (€125–230 million per year) facing the highest projected costs (Table 4). Other countries with substantial costs of dyke reinforcement are Denmark, Ireland, Spain and the Netherlands (>€50 million per year under all scenarios), as well as Sweden, Poland, Greece, Portugal and Belgium (all above €20 million per year under all scenarios).

Considering only the locations where further protection is needed, the additional average coastal defence height needed in Europe is 92 and 104 cm under sustainability and fossil fuel based

development, respectively (Table 4). Country average values vary from a minimum of 31–39 cm (Malta) to a maximum of 2.85–3.43 m (Belgium), with the range expressing the uncertainty among scenarios. Apart from Belgium, other countries with additional protection height above the EU mean are Slovenia (2.12–2.32 m), Poland (1.57–1.66 m), UK (1.47–1.5 m), Germany (1.42–1.44 m), the Netherlands (1.30–1.53 m), Latvia (0.83–1.35 m), and Estonia (0.97 and 1.42 m).

**Damages after applying additional protection**. Applying adaptation that optimizes NPV of coastal protection everywhere along the European coastline would still result in losses from coastal flooding, especially towards the end of the century. By 2100, EU total EAD will reach €8.8 and €24 billion under Sustainability and Fossil Fuel Based Development, respectively (Table 5). However, these represent a 96% (€200.1 billion) and 98% (€1.24 trillion) reduction compared to a do-nothing scenario. The highest losses are projected for Scotland, Ireland, Denmark, Romania, Croatia, Cyprus, Sicily, Andalucía, Bretagne, the south east Baltic Sea, and Provence, with NUTS2 level EAD exceeding €300 million towards the end of the century under Fossil Fuel Development (Table 5).

Similarly, further coastal protection will still result in people flooded with the EU total expected annual number of people flooded (EAPF) towards the end of the century reaching 653.4 k and 1343.1 k under Sustainability and Fossil Fuel Based Development, respectively (Table 5). Still the additional protection will reduce the 2100 EAPF along the entire European coastline by 59%

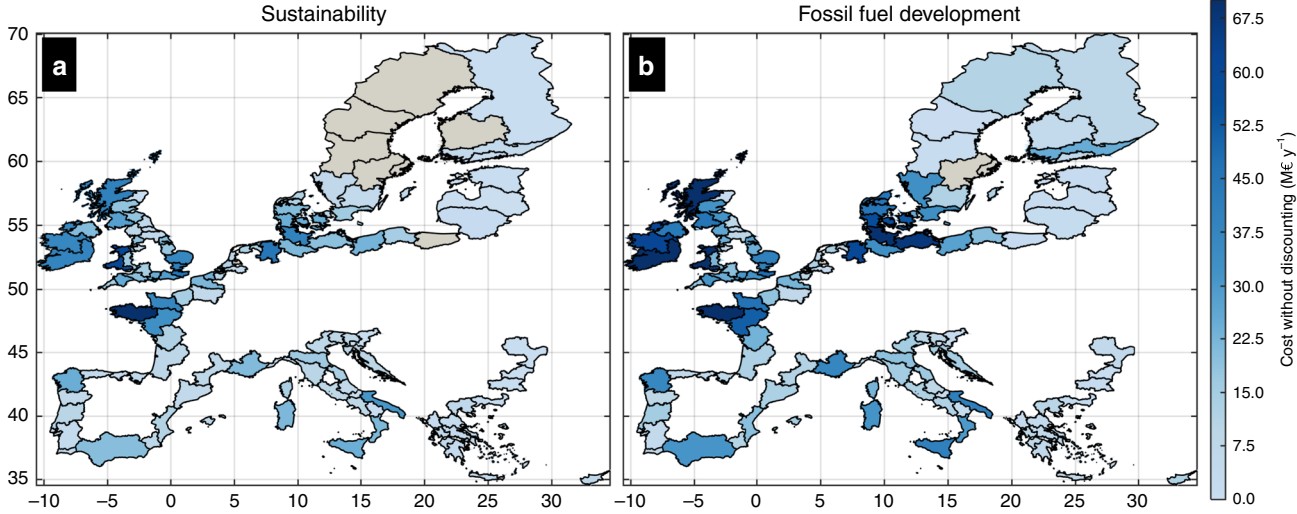

**Fig. 3 Cost of coastal protection at regional level.** Expected annual undiscounted costs of adaptation per NUTS2 region (expressed in million € per year) for Sustainability with RCP4.5 (**a**) and fossil fuel based development with RCP8.5 (**b**). Colours express the expected annual value averaged over the entire century (2020–2100).

**Table 4 Additional protection and related costs.**

|  | Costs (million €) | | Protection height increase (m) | |
| --- | --- | --- | --- | --- |
|  | **Sustainability** | **Fossil fuel based development** | **Sustainability** | **Fossil fuel based development** |
| Belgium (BE) | 32.89 | 31.97 | 3.43 | 2.85 |
| Bulgaria (BG) | 1.13 | 2.44 | 0.59 | 0.70 |
| Cyprus (CY) | 8.30 | 12.57 | 0.80 | 0.97 |
| Germany (DE) | 125.51 | 229.64 | 1.42 | 1.44 |
| Denmark (DK) | 90.20 | 224.06 | 0.88 | 1.00 |
| Estonia (EE) | 0.77 | 1.50 | 0.97 | 1.42 |
| Spain (ES) | 93.12 | 148.73 | 0.61 | 0.82 |
| Finland (FI) | 6.18 | 42.99 | 0.75 | 0.79 |
| France (FR) | 269.72 | 385.01 | 0.93 | 1.12 |
| Greece (GR) | 46.20 | 64.96 | 0.65 | 0.78 |
| Croatia (HR) | 8.86 | 14.37 | 0.50 | 0.64 |
| Ireland (IE) | 75.48 | 135.29 | 0.90 | 1.02 |
| Italy (IT) | 180.28 | 260.92 | 0.72 | 0.92 |
| Lithuania (LT) | 1.65 | 2.52 | 0.99 | 1.23 |
| Latvia (LV) | 0.32 | 0.92 | 0.83 | 1.35 |
| Malta (MT) | 0.16 | 0.48 | 0.31 | 0.39 |
| The Netherlands (NL) | 64.67 | 56.35 | 1.53 | 1.30 |
| Norway (NO) | 125.71 | 296.18 | 0.63 | 0.78 |
| Poland (PL) | 37.72 | 49.67 | 1.57 | 1.66 |
| Portugal (PT) | 26.22 | 37.03 | 0.95 | 1.04 |
| Romania (RO) | 0.76 | 6.12 | 0.62 | 0.87 |
| Sweden (SE) | 27.37 | 91.55 | 0.67 | 0.89 |
| Slovenia (SI) | 8.68 | 9.33 | 2.12 | 2.32 |
| United Kingdom (UK) | 522.65 | 719.15 | 1.47 | 1.50 |
| Total | 1754.55 | 2823.76 | 0.92 | 1.04 |

Annual mean costs of raising the dykes per country after discounting (million €), and corresponding mean, country-level increase in dyke height (m), under sustainability with RCP4.5 and fossil fuel based development with RCP8.5.

(959.2 k people) and 66% (2555.1 k), respectively, compared to a 'do nothing' scenario (Table 5). The population around the Puglia, Croatia, the Ionian Islands, the Basque country, Basse Normandie, Nord Pas de Calais, Scotland, Ireland and south east UK are projected to be more affected by coastal floods, with NUTS2 level EAPF exceeding 15,000 towards the end of the century under Fossil Fuel Based Development (Table 5).

**Discussion**. The present analysis has shortcomings that are inherent to the scale of the application. We note that in our cost-benefit analysis the benefits are limited to avoided flood risk until the end of the 21st century. Other potential costs of increased protection of coastal areas against inundation, such as the loss of valuable ecosystems through coastal squeeze[26,27], are not included in the analysis.

**Table 5 Damages after adaptation.**

| | EAD2100 adapt | | EAPF2100 adapt | |
|---|---|---|---|---|
| | Sustainability | Fossil fuel based development | Sustainability | Fossil fuel based development |
| Belgium (BE) | 0.03 | 0.03 | 13.10 | 29.21 |
| Bulgaria (BG) | 0.01 | 0.06 | 0.60 | 0.73 |
| Cyprus (CY) | 0.05 | 0.21 | 8.06 | 8.63 |
| Germany (DE) | 0.69 | 2.05 | 16.76 | 55.13 |
| Denmark (DK) | 1.14 | 2.82 | 26.42 | 54.45 |
| Estonia (EE) | 0.02 | 0.07 | 0.14 | 0.30 |
| Spain (ES) | 0.35 | 0.96 | 73.78 | 100.99 |
| Finland (FI) | 0.08 | 0.65 | 1.85 | 10.30 |
| France (FR) | 0.89 | 2.40 | 55.88 | 140.92 |
| Greece (GR) | 0.42 | 1.33 | 30.06 | 55.04 |
| Croatia (HR) | 0.13 | 0.39 | 10.72 | 14.59 |
| Ireland (IE) | 0.61 | 1.24 | 41.56 | 83.02 |
| Italy (IT) | 0.68 | 1.95 | 44.69 | 69.31 |
| Lithuania (LT) | 0.02 | 0.05 | 1.49 | 2.46 |
| Latvia (LV) | 0.01 | 0.02 | 0.15 | 0.31 |
| Malta (MT) | 0.00 | 0.01 | 0.09 | 0.12 |
| The Netherlands (NL) | 0.23 | 0.87 | 4.94 | 7.90 |
| Norway (NO) | 1.24 | 2.74 | 42.03 | 126.48 |
| Poland (PL) | 0.07 | 0.32 | 12.43 | 18.20 |
| Portugal (PT) | 0.12 | 0.34 | 5.81 | 9.25 |
| Romania (RO) | 0.02 | 0.65 | 0.85 | 1.24 |
| Sweden (SE) | 0.31 | 1.21 | 9.42 | 18.35 |
| Slovenia (SI) | 0.01 | 0.02 | 0.25 | 0.38 |
| United Kingdom (UK) | 1.77 | 3.60 | 252.30 | 535.73 |
| Total | 8.88 | 23.98 | 653.39 | 1343.05 |

Expected annual damage (EAD) and expected annual number of people flooded) from coastal flooding after implementing additional protection (billion €), under sustainability with RCP4.5 and fossil fuel based development with RCP8.5.

Near river deltas and estuaries coastal and river flooding could coincide. Such compound events could reinforce each other and give rise to impacts that are larger than the sum of the impacts of the single events. With rising extreme sea levels along Europe's coastlines and increasing river flood hazard in many parts of Europe, compound flood hazard will likely increase in Europe[28,29]. A proper assessment of this hazard and the consequent risk is yet lacking at continental scale, it should be noted, however, that to date compound flood risk represents only a marginal fraction of the total flood risk in Europe[30].

Sea levels are projected to increase long after 2100 and very likely this will happen at an accelerating rate[31,32]. Hence, even though our impact and cost-benefit analysis is limited until 2100, adaptive measures taken now will also lower flood risk during the 22nd century and beyond. Considering longer time spans, the benefits and maintenance costs of rising dyke heights are therefore likely much higher than estimated in this paper.

The results of this paper are difficult to compare with those of earlier studies due to differences in the methodology and scenarios applied. The dedicated European coastal flood risk studies of Hinkel et al.[33] and PESETA II[34] were still based on the Fourth Assessment Report[35] of the Intergovernmental Panel on Climate Change (IPCC), using lower 21st century sea-level rise scenarios and, therefore, projecting lowed coastal flood damages. Furthermore, none of these studies has applied a cost and benefit analysis (CBA) approach to coastal protection. More recently, LT18[36] applied CBA to assess coastal protection scenarios at the global scale, but using a methodology which differs from the present one in several aspects: (i) LT18 follow a different definition of ESLs compared to the present study, omitting wave contributions, spatial dependencies and the effects of changing atmospheric conditions throughout the century; (ii) L18 applied a bathtub approach, compared to a hydrodynamic model used here

for flood propagation; (iii) spatial variations in economic exposure enter the analysis of L18 at a higher resolution (i.e. census units of gridded population data) as compared to our study (i.e. NUTS2 level); and (iv) our study uses a higher thematic resolution of types of economic exposure (i.e. land use classes). Despite the above the results are similar. For the EU28, L18 find a baseline (i.e. 2020) EAD of US$ 1.5–1.7 billion per year, which is close to the €1.4 billion per year baseline of this study, and a residual EAD in 2100 of US$ 10 billion per year under a 52 cm global mean SLR scenario (SSP1-RCP2.6 high ice melt) which is also close to the €9 billion per year, presently reported under our comparable Sustainability scenario (57 cm mean SLR). Finally, our study comes in line with the latest IPCC report on the Ocean and the Cryosphere which reports that coastal protection can reduce the EAD in 2100 by 2–3 orders of magnitude[20].

Our results highlight potential savings in terms of avoided damages that can be obtained through strategies that increase structural flood protection at the subnational scale. Large-scale studies as the one presented help to initiate the dialogue with stakeholders and identify priority regions. For the actual implementation of individual measures at local scale, they should be complemented by detailed studies that are performed using local models and data. Local studies are also more appropriate to evaluate the co-existence of hard protection with other adaptation practices, such as nature-based solutions, retreat or accommodate. This requires very detailed planning using local data not publicly available at European scale. By no means we suggest that hard protection is the only or preferable option, and our analysis does not exclude the parallel implementation of more sustainable environmental practices to enforce the physical and ecological resilience of coastal zones[20]. Hard protection has been the most common approach until now and for that reason we consider the present analysis useful for practitioners and policy makers.

**Table 6 The effect of discounting.**

|                                      | Sustainability | Fossil fuel based development |
|--------------------------------------|----------------|-------------------------------|
| Protection height increase (M)       | 0.04           | 0.11                          |
| EAD2100 adapt decrease               | 5.11           | 16.43                         |
| EAPF2100 adapt decrease              | 118.90         | 40.58                         |
| Mean country level BCR increase      | 6.96           | 17.98                         |
| % Coastline BCR > 1 increase         | 7.53           | 7.76                          |
| Costs (billion €) increase           | 1.41           | 2.21                          |

Comparing results obtained with and without discounting for the entire European coastline, under sustainability with RCP4.5 and fossil fuel based development with RCP8.5: reduction in expected annual damage (EAD); Expected annual number of people flooded) from coastal flooding after implementing additional protection (billion €); Annual mean costs of raising the dykes per country after discounting (million €); corresponding mean, country-level increase in dyke height (m); percentage of the country's coastline with mean BCR > 1 (benefits of adaptation exceed the costs); and mean country level benefit to cost ratio (BCR).

The applied probabilistic framework allows decision makers to interpret the results according to the amount of risk they consider as acceptable. Our projections of future coastal hazard and risk, as well as dyke costs, come with uncertainty, and in this paper we evaluated the adaptation option that optimizes the expected benefits vs costs. However, stakeholders could select a more conservative criterion and optimize adaptation investments in view of high-end, less probable future scenarios, under which flood impacts will be higher, instead of the whole range currently considered. Such a choice would result in higher adaptation costs, but would also imply lower risks for future generations, as the analysis would prioritize protection against the rarer and more catastrophic events.

The use of discounting is another critical aspect as high discount rates put more weight on short term costs and benefits. Here this means that capital investment costs in protection, which occur now, are emphasized, while future benefits of adaptation are downgraded. In sum this may discourage taking action now. For the same analysis without discounting the estimated dyke height is on average 4–11 cm higher along the European coastline (Table 6), compared to when discounting costs and benefits. This would require an additional mean annual cost of €1.41 billion and €2.21 billion during this century, depending on the scenario. The benefits of the increased protection would be that an additional 7.5%–9.7% of Europe's coastline would be protected to rising seas, and the EAD and EAPF by the end of the century would be further reduced by €5.1–16.4 billion and 40–255 thousand people, respectively, compared to when not discounting.

## Methods
**Coastal flood risk and adaptation modelling framework**. The coastal flood risk analysis is based on the model LISCOAST (large-scale integrated sea-level and coastal assessment tool). The modular framework has been developed to assess weather-related impacts in coastal areas in present and future climates. It combines state-of-the-art large-scale modelling tools and datasets to quantify hazard, exposure and vulnerability and compute consequent risks[10]. The modelling framework was further extended to evaluate costs and benefits of heightening dyke heights and find the optimal adaptation design based on maximizing the net present value. More details on the different steps of the analysis are provide below.

**Hazard assessment**. Coastal flood impacts are driven by nearshore extreme sea levels (ESLs). In this study, ESLs are modelled along the European coastline using segments of variable length with a maximum of 25 km for the most straight coastline stretches. Our projections go to the end of the 21st century and we consider Representative concentration pathways (RCPs) RCP4.5 and RCP8.5, for which an ensemble of 6 climate models have been used to account for uncertainty in climate projections. RCP4.5 can be considered as a moderate emissions-mitigation-policy scenario and RCP8.5 as a high-emissions scenario. ESLs are calculated by adding linearly the contributions of different components:

$$ESL = SLR + \eta_{CE} + \eta_{tide} \qquad (1)$$

where

$SLR$ is the sea level rise, obtained from a GCM ensemble combined with contributions from ice-sheets and ice-caps[37].

$\eta_{CE}$ is the contribution from extreme wind and atmospheric pressure, driving waves and storm surge, that is obtained dynamic ocean simulations[37,38].

$\eta_{tide}$ the maximum tidal level sampled probabilistically to express the spring-neap variation of the high tide water level.

We then apply in each coastal segment non-stationary extreme value analysis[39] to the ESL projections. From the fitted extreme value distributions, we obtain ESLs for a range of return periods (inverse of probability) between 2 and 20,000 years. Hence, ESLs are expressed as a function of time and return period[22]

$$ESL = f(\text{year}, \text{RP}) \qquad (2)$$

The ESLs in Eq. (2) are subsequently used as forcing for coastal flood inundation calculations at 100 m resolution using the hydrological model Lisflood-FP[40]. The inundation simulations take into account present coastal protection standards which were derived from the FLOPROS dataset[41], combined with available regional/national reports and datasets (for more details see Vousdoukas et al.[10]). Land surface elevation data are provided from the Shuttle Radar Topography Mission (SRTM) DEM[42]. This results in time-varying coastal flood inundation maps for each of the considered return periods and for each coastal segment. The ESLs do not consider non-linear interactions between SLR, tides, storm surge and waves. This is an inevitable assumption given the spatio-temporal scale of the analysis, which is common in similar studies[15,36,43,44] and is considered to have an effect that is by far lower that of the other uncertainty factors[45,46].

**Exposure and vulnerability**. The resulting flood inundation maps are combined with exposure and vulnerability information at the corresponding point in time in order to estimate direct flood damages[10,47,48]. Baseline exposure (reference year 2012) is available from the refined CORINE land use/land cover dataset (CLC) at 100 m resolution, featuring 44 different land use classes[49]. Baseline population maps (reference year 2011) are available from Batista e Silva et al.[49]. For future population exposure we used global projections gridded at 1/8° resolution of population density and urban population[1] based on the respective SSPs. The gridded projections of urban population were considered as a proxy of the degree of urbanization. Given that urban land use classes contribute to >90% of the estimated damages, relative changes in urbanization were used to estimate changes in damages due to land use change. Consistent country level GDP projections under SSP1, SSP3 and SSP5 are available from IIASA and OECD[50,51]. The projected changes in country-level GDP from both sources were spatially distributed according to the patterns of change in the gridded projections of population. Asset values for future time slices were adjusted by scaling per NUTS3 region the depth damage functions according to changes in the future NUTS3 GDP per capita compared to the baseline. Given that the spatial resolution of the baseline population maps is lower than the projections, we obtain future population in each 100 m grid point by multiplying it with the relative change reported in the nearest grid point of the projections dataset.

RCPs and SSP are combined according to van Vuuren and Carter[23], who suggest that (i) RCP4.5 is compatible with global sustainable development (sustainability; SSP1); and (ii) RCP8.5 is compatible with socio-economic development driven by mitigation challenges (fossil fueled development: SSP5).

The vulnerability to coastal flooding of coastal infrastructure, societies and ecosystems is expressed through depth-damage functions (DDFs)[52,53]. DDFs define for each of the 44 land use classes of the refined CORINE Land Cover the relation between flood inundation depth and direct damage. The country-specific DDFs were further rescaled at NUTS3 level based on GDP per capita to account for differences in the spatial distribution of wealth within countries.

**Estimation of people flooded and direct losses**. For each coastal segment, people flooded and direct flood losses in time for the different return periods are calculated at 100 m resolution by combining the corresponding flood inundation maps with the exposed people and assets and the vulnerability functions. Areas that are inundated on a regular basis (which could happen in the future with sea level rise), here defined as the areas that lie below the high tide water level, are considered as fully damaged and the maximum loss according to the DDFs is applied. For areas inundated only

during extreme events, the damage is estimated by applying the DDFs combined with the simulated inundation depth for the respective return period events. For each coastal segment this results in annual estimates up to 2100 of coastal flood damage D (and people flooded) for the range of return periods considered

$$D = f(\text{year, RP}) \tag{3}$$

**Probabilistic projections of flood impacts**. Projections of future flood impacts are estimated in a probabilistic framework. For a correct statistical description of the hazard, it is necessary to consider spatial dependency in the occurrence of extreme events along the European coastline. If a severe storm hits a point along the coast, nearby locations will likely also be exposed to extreme conditions, and neglecting such dependency would lead to an underestimation of the aggregate risk. To that end, the spatial dependencies of ESLs were estimated through copulas. Considering the spatial dependencies among coastal segments, we produce 10,000 realizations of sequences of ESLs during the present century though Monte Carlo simulations. This produces annual time series of return periods (corresponding to the respective ESLs) for each coastal segment. The time series cover 80 years from 2020 until 2100, resulting in a $80 \times 10,000$ matrix of extreme event return periods ($RP_{matrix}$) for each segment, with dimensions corresponding to the number of years and Monte Carlo realizations, respectively. The matrices of return periods are transformed into matrices of direct losses ($D_{matrix}$) for each segment according to Eq. (3). The number of realizations was chosen after several preliminary tests during which it was shown that 10,000 ensured convergence both in terms of mean and standard deviation values (fluctuations below 0.001%).

**Estimation of adaptation costs**. In order to estimate the optimal dyke design for a coastal segment we consider dyke heights ($Z_{prot}$) that vary from the current level ($Z_{prot,pres}$) to a maximum elevation. The latter exceeds by 1 m the 99th ESL quantile estimated for that coastal segment during the present century ($ESL_{max}$). We discretize the range between $Z_{prot,pres}$ and $ESL_{max}$ in 40 increments. Hereby we assume that $Z_{prot,pres}$ is upgraded gradually to the desired design between 2020 and 2050, and remains constant until the end of the century.

Costs of dyke heightening are calculated by aggregating investment and maintenance costs during the entire study period. Investment costs are expressed as a linear function of dyke heightening; which has been shown to be a good approximation[54], and has been used in the previous studies[24,36,55]. Country estimates of investment costs of dykes are available from two sources: (i) the dataset used in the analysis of global investment costs for coastal defences of *Nicholls, Hinkel*[55]; and (ii) the dataset used in the global flood analysis of Ward et al.[24]. In both datasets costs are expressed as investment costs in US$ per metre heightening considering differences in construction costs across countries, which were converted to 2015 € values using GDP deflators and market exchange rates obtained from Eurostat. Maintenance costs are assumed to be 1% per year of capital investment costs[56]. The km length of dykes is equivalent to the coastline length of each segment that was derived from OpenStreetMaps. Dyke heights are assumed to be uniform over the entire segment. Both datasets on dyke unit costs come with confidence intervals, on the basis of which cost probability density functions were fitted. In the probabilistic framework costs are randomly sampled from these distributions assuming that each dataset has equal probability of occurrence.

**Estimation of adaptation benefits**. Benefits are represented here as the avoided damages by increasing the dyke height in a coastal segment. For each of the 40 increments, benefits are calculated for the 10,000 projections of ESL up to 2100 (Eq. 2) as the difference between future losses with and without additional coastal protection, aggregated over the entire study period. We assume that if the ESL of the event (Eq. 2) does not exceed the dyke height then the damage of that event will be zero. If the ESL overtops the dyke, then it breaches and the damage is obtained from Eq. (3). For each coastal segment. This results in a matrix of 40 increments vs 10,000 estimates of benefits.

**Cost-benefit analysis**. The objective of the cost-benefit analysis is to find the protection standard for each coastal segment that maximises the Net Present Value (NPV). The latter is the sum over the project lifetime of the costs and benefits associated with a specific investment and determines whether a project will deliver sufficient benefits to justify the costs. We, therefore, sample 10,000 realizations of unit cost from the cost distributions, which are used to generate 10,000 estimates of the (capital and maintenance) costs for each of the 40 increments in a coastal segment. These are combined with the 10,000 realizations of benefits for each increment, and the NPV is calculated. This results in 10,000 NPVs for each dyke increment. This allows to derive the probability that a certain dyke design is cost-effective, and central estimates for each increment can be used to choose an optimal dyke height. Here we use the mean NPV of the 10,000 NPVs to choose the optimal design. For the dyke elevation that maximises the mean NPV in a coastal segment, we also calculate the benefit-cost ratio (BCR), which is the ratio of its total benefits to its total costs. Results at larger scales (e.g., NUTS2, country, or EU-level) are obtained by summing NPVs estimates over the coastal segments in the area of interest.

The benefits delivered by dykes often occur long after they have been constructed. Discounting is used to reflect that the costs and benefits incurred in the future are of less value than those delivered in the near term. In order to determine the present value of future costs and benefits, they are discounted and aggregated according to:

$$X_{\text{present}} = \sum_{t=1}^{T} \frac{X_t}{(1 + r_{sw})^t} \tag{4}$$

where $T$ represents the duration of the project's lifetime in years, $X_t$ is the cost or benefit incurred over a year by the project and $r_{sw}$ is the social welfare discount rate. The choice of the latter can largely influence the cost-benefit analysis of the adaptation measure. Larger values of $r_{sw}$ tend to discourage the implementation of the policy, as discounted future benefits of the measure become smaller compared to its costs that are incurred earlier in time. We note that we limit $T$ by putting 2100 as the end of the project lifetime, yet in reality the lifetime of the dykes is likely longer.

We calculate the social discount rate using the Ramsey equation[57], which combines information about the growth of the economy with two main parameters: the rate of pure time preference of society and the elasticity of the marginal utility of consumption. The formula is:

$$r_{sw} = \rho + \eta \cdot g \tag{5}$$

where:
$\rho$ is the rate of pure time preference;
$g$ is the growth rate of per capita consumption;
$\eta$ is the elasticity of the marginal utility of consumption.

The Ramsey equation reflects the two main reasons why the society or a hypothetical social planner would discount future costs and benefits. A value of $\rho$ larger than zero reflects impatience and a preference for consumption in the current period rather than consumption in the future. On the contrary if $\rho$ is equal to zero, the society has no preference for a unit of consumption today or in the future. It is often referred to as the inter-generational equity parameter, as it reflects preferences between the present and future generations. The other reason why future costs and benefits are discounted is the decreasing marginal utility of consumption ($\eta$) and the growth rate of per capita income ($g$). A high value of $\eta$ reflects a fast declining utility as consumption increases. The interpretation is that the wealthier the society, the lower the utility derived from an equal increase of the consumption level; therefore with a positive $g$, future benefits will have a lower value in the present.

Here the Ramsey equation is calibrated using average growth rates for consumption per capita. As suggested in the EC Guide to Cost-Benefit Analysis of Investment Projects[58], we distinguish between the so-called Cohesion countries, which benefit from the Cohesion Fund, and the rest of the EU Member States[59]. For the 2014-2020 period, the Cohesion Fund concerns Bulgaria, Croatia, Cyprus, the Czech Republic, Estonia, Greece, Hungary, Latvia, Lithuania, Malta, Poland, Portugal, Romania, Slovakia and Slovenia. From our macroeconomic projections the average growth rate of consumption per capita for cohesion countries is equal to 2%, while for the rest of the countries is 1% per year.

We further assume a value of 1 for $\rho$, which is chosen as a central value between 0, i.e. no preference for current or future generations, and 2, which is the value attributed by Weitzman in his review of the Stern Review[60]. Values of $\eta$ in the literature typically range between 1 and 4, with a central estimate of 2[61], which is the value that we assume here.

With these values for $g$, $\rho$, $\eta$, the resulting discount rates are 5% the cohesion countries (poorer countries in Europe) and 3% for the other Member States. The resulting discount rates appear to be in line with those suggested by the European Commission for the cost-benefit analysis of major investment projects[60,61].

**Reporting summary**. Further information on research design is available in the Nature Research Reporting Summary linked to this article.

## Data availability
The models and datasets presented are part of the integrated risk assessment tool LISCoAsT (Large scale Integrated Sea-level and Coastal Assessment Tool) developed by the Joint Research Centre of the European Commission. The dataset is available through the LISCoAsT repository of the JRC data collection: https://doi.org/10.2905/D3163B12-D931-44ED-A609-AF82860A47E0 PID: http://data.europa.eu/89h/d3163b12-d931-44ed-a609-af82860a47e0.

## Code availability
The code that supported the findings of this study is available from the corresponding author upon reasonable request.

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

## Acknowledgements

P.J.W. received additional funding from the Dutch Research Council (NWO) in the form of a VIDI grant (grant no. 016.161.324). J.H. has received funding from the European Union's Horizon 2020 Programme under the COACCH project (Grant agreement no. 776479) and from the German Federal Ministry of Education and Research through the ERA4CS project ISIPEDIA (Grant no. 01LS1711C).

## Author contributions

M.I.V. and L.F. jointly conceived the study. M.I.V. and L.M. produced the extreme sea level datasets. P.W. and J.H. produced the unit cost estimates for dyke height increase. M.I.V., L.F., I.M. and J.C.C. estimated the direct losses from coastal flooding. M.I.V. analysed the data and prepared the paper, with all authors discussing results and implications and commenting on the paper at all stages.

## Competing interests

The authors declare no competing interests.
