## [Peer Review File · Nature Communications]

Reviewers' comments:

Reviewer #1 (Remarks to the Author):

The manuscript presents an analysis of the potential impacts of extreme sea levels (ESL) and associated flooding in Europe and assesses the benefits of hard protection, under a range of physical and socioeconomic scenarios. This is an interesting follow up to previous studies and goes beyond previous work by employing a fully probabilistic framework that uses dynamic simulations of all future ESL components and inundation for assessing impacts. Importantly, the analysis accounts for the spatial dependencies of ESLs along the European coastline, which to my knowledge has not been the case in any previous impact assessment studies (which the authors have been rather modest about and could possibly emphasise in their manuscript). The results provide useful insights into future adaptation needs and benefits and can be used to support the development of future adaptation strategies at the EU level. The manuscript is well written and the analysis is technically sound. I therefore believe that the paper is, in principle, suitable for publication. I do however have a series of comments and important questions on methods, which I list below and I believe the authors should address before final acceptance of the manuscript.

- The authors have employed a combination of SSP3 and what they refer to as a high emissions scenario (which I assume is the RCP8.5), based on Van Vuuren and Carter (2014). Recent work has shown that this is not a plausible combination, as slow economic development in SSP3 cannot lead to the high forcing levels of RCP8.5 (see Rogelj et al. 2018, suppl. Material, in Nature Climate Change). The only SSP that can be combined with RCP 8.5 is SSP5 (and even this has been questioned recently). If the authors insist on using the combination of SSP3 and RCP8.5 they would need a very strong justification. Alternatively I would assume that the authors may have already conducted runs for SSP3 and RCP6.0, which they could present instead.
- Following on the previous comment, there have been explicit recommendations from the scenarios community (also stemming from the above discussion) to avoid the "business as usual" term (line 406) as there is no business as usual in s-e development (which is exactly the reason why we use scenarios). I would therefore suggest the authors to remove this term, which also does not offer much to the sentence anyhow.
- The manuscript does not provide any evaluation of the results. If I am not mistaken there have been previous studies that have assessed impacts of sea level rise at the European level (e.g. PESETA; or the study of Hinkel et al. 2010, in Adaptation and Mitigation Strategies for Global Change). They do use different scenarios (SRES) but these can be easily "mapped" onto the SSP/RCP combinations and possibly provide some useful comparisons.
- I am a little confused as to why the authors have used a baseline population map that has been developed with different methods and input base data (also formats) from the future population maps. I do not have access to the Batista Silva et al. paper that the authors cite, however, based on the abstract this is a land use study and not a population study. The combination of these different data sources could introduce significant errors/biases in the analysis and this is something that the authors should clarify.
- Further to the previous point, the authors have used the coarse resolution data of Jones and O'Neill 2017 (which are much coarser than the other data that the authors have used) whereas there exist 1-km resolution data available online. What was the reason for this?
- If I understand correctly, the calculation of ESLs does not account for tide-surge interactions? If this is the case, I think that the authors should clearly state it.
- When calculating present day losses, do the authors consider current protection measures or protection standards? This point was not clear to me and it could be useful to clarify in the manuscript.
- The authors have used a discount factor of 5. I am not an economist but from what I have seen, previous studies on assessing the impacts of climate change have mostly used lower discount factors.

Also, to my knowledge, results can be very sensitive to the choice of the discount factor. Could the authors justify the use of 5?

- From this type of work I would certainly expect some discussion on the implications of those results in terms of policy. I believe such a discussion (even if it only were a couple of lines long) would certainly strengthen the manuscript.

- The manuscript leaves the impression that hard protection is a clear no-brain solution to rising ESLs because it is generally economically beneficial. However, as the authors correctly note in the introduction, hard protection often results in other problems. It is also not always desirable (e.g. in tourist areas, for example in the Mediterranean). Further, decisions are not only made on monetary considerations but, for example, social costs are also considered. The discussion on these points is very limited in the manuscript and may communicate the wrong message to the readers.

- I was a bit surprised to see rather high CB ratios in poorer, not very densely populated coasts (e.g. south Italy). I imagine that these may result from e.g. large cities that are located in those administrative units? No need to address this in the manuscript, just a question from my side.

- Just a suggestion: I would not use the word "incentive" in the title as "incentive" usually implies receiving something in return. I would opt for "benefits" or "motivation". But this is up to the authors and editor to decide

Finally, two typos that I noticed:

- line 60: i find the wording in point (v) confusing, it reads as if the authors also calculate river flood damages. Maybe replace "follow" with "following"?

- Table 2, legend: "people flooded ... to coastal flooding" - should probably be "exposed"?

I hope that the authors find my comments useful for improving the manuscript.

Reviewer #2 (Remarks to the Author):

This manuscript is good piece of work an it can be suitable for publication after the solution of two specific aspects:

1) I found the same manuscript (from the same authors) online published in the folowing web page:

<https://eartharxiv.org/wu4r6/>

No idea about how this works (is this possible?), but my major concern is a possible duplicity. Why duplicity? : that manuscript or preprint has a doi: 10.31223/osf.io/wu4r6

If Nature communications accept this manuscript, then I think we have the same manuscript published two times.

2) Along thefirst part of the manuscript were presented some "adaptation" measures. These measures are not really adaptation measures. Some of them are protection measures. I suggest a deep update about the existing management strategies, here some help.

Protection

This approach consists in the preservation of vulnerable areas (i.e., population centers, economic activities and resources) by means of constructing hard structures, or alternatively soft protection measures which does include beach restoration. Both hard and soft approaches seek to eliminate or mitigate damage to the coast by use of barriers to reduce wave impact and control hydraulic processes that influence sediment transport.

Accommodation

Accommodation includes all approaches for addressing flooding by revising and reorganizing human activities in the coastal zone, or even at the river basin level. This approach allows acceptance of a certain degree of flooding and erosion by construction methods, changing land use, and improving preparedness. The above means a continued usage of land at risk without attempting to prevent the area from being damaged by natural events, and allowing conservation and migration of ecosystems.

Use of ecosystems

This approach is based on the natural environmental influences over processes related to flooding (e.g., sediment capture and energy attenuation) by restoring or creating ecosystems. The use of ecosystems for flooding management is an integrated process to conserve and improve ecosystem health that sustains ecosystem services for human well-being.

Managed/planned retreat

Managed/planned retreat has the primary objective of reducing population, property, and infrastructure at risk through the planned withdrawal from coastal hazard zones, including flood-prone areas. Specific to the flooding hazard, managed retreat implements mitigation tools designed to move existing and proposed development out of the path of both short- and long-term coastal flooding. The approach may include managed realignment where shore-protection structures are removed selectively to allow natural coastal environments to be re-established.

Regards,

Response to reviewers' comments

Reviewer 1:

The manuscript presents an analysis of the potential impacts of extreme sea levels (ESL) and associated flooding in Europe and assesses the benefits of hard protection, under a range of physical and socioeconomic scenarios. This is an interesting follow up to previous studies and goes beyond previous work by employing a fully probabilistic framework that uses dynamic simulations of all future ESL components and inundation for assessing impacts. Importantly, the analysis accounts for the spatial dependencies of ESLs along the European coastline, which to my knowledge has not been the case in any previous impact assessment studies (which the authors have been rather modest about and could possibly emphasise in their manuscript). The results provide useful insights into future adaptation needs and benefits and can be used to support the development of future adaptation strategies at the EU level. The manuscript is well written and the analysis is technically sound. I therefore believe that the paper is, in principle, suitable for publication. I do however have a series of comments and important questions on methods, which I list below and I believe the authors should address before final acceptance of the manuscript.

AUTHORS: We would like to thank the reviewer for the positive feedback. We have also added a sentence about the spatial dependencies in the beginning of the fifth paragraph:

‘One of the important advances of the present study is that it resolves the spatial dependencies of ESLs along the European coastline, considering temporally dynamic contributions not only from SLR, but also astronomical and meteorological components. ‘

- The authors have employed a combination of SSP3 and what they refer to as a high emissions scenario (which I assume is the RCP8.5), based on Van Vuuren and Carter (2014). Recent work has shown that this is not a plausible combination, as slow economic development in SSP3 cannot lead to the high forcing levels of RCP8.5 (see Rogelj et al. 2018, suppl. Material, in Nature Climate Change). The only SSP that can be combined with RCP 8.5 is SSP5 (and even this has been questioned recently). If the authors insist on using the combination of SSP3 and RCP8.5 they would need a very strong justification. Alternatively I would assume that the authors may have already conducted runs for SSP3 and RCP6.0, which they could present instead.

AUTHORS: We would like to thank the reviewer for this valid point and we have removed the combination of RCP8.5 and SSP3 from our analysis.

- Following on the previous comment, there have been explicit recommendations from the scenarios community (also stemming from the above discussion) to avoid the "business as usual" term (line 406) as there is no business as usual in s-e development (which is exactly the reason why we use scenarios). I would therefore suggest the authors to remove this term, which also does not offer much to the sentence anyhow.

AUTHORS: We have removed 'business as usual' everywhere in the document

- The manuscript does not provide any evaluation of the results. If I am not mistaken there have been previous studies that have assessed impacts of sea level rise at the European level (e.g. PESETA; or the study of Hinkel et al. 2010, in Adaptation and Mitigation Strategies for Global Change). They do use different scenarios (SRES) but these can be easily "mapped" onto the SSP/RCP combinations and possibly provide some useful comparisons.

AUTHORS: We have added a section discussing the present findings against the ones of previous studies:

‘The results of this paper are difficult to compare with those of earlier studies due to differences in the methodology and scenarios applied. The dedicated European coastal flood risk studies of Hinkel et al.¹ and PESETA II² were still based on the Fourth Assessment Report³ of the Intergovernmental Panel on Climate Change (IPCC), using lower 21st century sea-level rise scenarios and therefore projecting lowered coastal flood damages. Furthermore, none of these studies has applied a cost and benefit analysis (CBA) approach to coastal protection. More recently, LT18⁴ applied CBA to assess coastal protection scenarios at the global scale, but using a methodology which differs from the present one in several aspects: (i) the present studies considers spatial and temporal dynamics of meteorological tides, not considered in L18; (ii) L18 applied a bathtub approach, compared to a hydrodynamic model used here for flood propagation; (iii) spatial variations in economic exposure enter the analysis of L18 at a higher resolution (i.e. census units of gridded population data) as compared to our study (i.e. NUTS2 level); and (iv) our study uses a higher thematic resolution of types of economic exposure (i.e. land use classes). Despite the above the results are similar. For the EU28, L18 find a baseline (i.e. 2020) EAD of US\$ 1.5-1.7 billion/year, which is close to the EUR 1.4 bn/yr baseline of this study, and a residual EAD in 2100 of US\$ 10bn/yr under a 52 cm global mean SLR scenario (SSP1-RCP2.6 high ice melt) which is also close to the € 9bn/yr, presently reported under our comparable Sustainability scenario (57 cm mean SLR). Finally, our study comes in line with the latest IPCC report on the Ocean and the Cryosphere which reports that coastal protection can reduce the EAD in 2100 by 2-3 orders of magnitude⁵.’

- I am a little confused as to why the authors have used a baseline population map that has been developed with different methods and input base data (also formats) from the future population maps. I do not have access to the Batista Silva et al. paper that the authors cite, however, based on the abstract this is a land use study and not a population study. The combination of these different data sources could introduce significant errors/biases in the analysis and this is something that the authors should clarify.

AUTHORS: We have high detail baseline exposure maps and we use the projected relative changes of population from the socio-economic projections to adapt them for the future. We have avoided describing the procedure in high detail as this is documented in our 2018 NCLIM paper. However we agree with the reviewer that this was not clearly explained in the original manuscript, and we have clarified the methodology as follows in the revision (see Exposure and Vulnerability in the Methods section):

‘The resulting flood inundation maps are combined with exposure and vulnerability information at the corresponding point in time in order to estimate direct flood damages⁶⁻⁸. Baseline exposure (reference year 2012) is available from the refined CORINE land use/land cover dataset (CLC) at 100 m resolution, featuring 44 different land use classes⁹. Baseline population maps (reference year 2011) are available from Batista e Silva et al⁹. For future population exposure we used global projections gridded at 1/8⁰ resolution of population density and urban population¹⁰ based on the respective SSPs. The gridded projections of urban population were considered as a proxy of the degree of urbanization. Given that urban land use classes contribute to >90% of the estimated damages, relative changes in urbanization were used to estimate changes in damages due to land use change. Consistent country level GDP projections under SSP1, SSP3 and SSP5 are available from IIASA and OECD^{11,12}. The projected changes in country-level GDP from both sources were spatially distributed according to the patterns of change in the gridded projections of population. Asset values for future time slices were adjusted by scaling per NUTS3 region the depth damage functions according to changes in the future NUTS3 GDP per capita compared to the baseline. Given that the spatial resolution of the baseline population maps is lower than the projections, we obtain future population in each 100 m grid point by multiplying it with the relative change reported in the nearest grid point of the projections dataset.’

- Further to the previous point, the authors have used the coarse resolution data of Jones and O'Neill 2017 (which are much coarser than the other data that the authors have used) whereas there exist 1-km resolution data available online. What was the reason for this?

AUTHORS: *We assume that the reviewer is referring to this dataset:*

<http://www.cgd.ucar.edu/iam/modeling/spatial-population-scenarios.html>. These data were not available at the time when our impact assessment was carried out⁷. We now performed some tests with the higher resolution dataset and we concluded that this has a limited effect on the projected impacts, mainly because the lower resolution dataset of Jones and O'Neil represents well inland migration patterns towards the coastal zone, which are responsible for a rise in coastal flood losses. Given the time and resources that it would take to re-run the whole analysis with the new dataset and the limited effect it has on the projected losses, we decided not to re-run with the new data. We are confident that the main findings on the costs/benefits of adaptation would not change because of the higher resolution in these projections.

- If I understand correctly, the calculation of ESLs does not account for tide-surge interactions? If this is the case, I think that the authors should clearly state it.

AUTHORS: *This is correct and it is now highlighted in the revision in the last sentence of the 'Hazard assessment' section in the methods:*

'The ESLs do not consider non-linear interactions between SLR, tides, storm surge and waves. This is an inevitable assumption but given the spatio-temporal scale of the analysis, which is common in similar studies^{4,13-15} and is considered to have an effect that is by far lower than that of the other uncertainty factors^{16,17}.

- When calculating present day losses, do the authors consider current protection measures or protection standards? This point was not clear to me and it could be useful to clarify in the manuscript.

AUTHORS: *We do consider existing protection, this information is available in the 'Hazard assessment' section and has been made even clearer in the revision:*

'The inundation simulations take into account present coastal protection standards which were derived from the FLOPROS dataset¹⁸, combined with available regional/national reports and datasets (for more details see Vousdoukas et al.⁷).

- The authors have used a discount factor of 5. I am not an economist but from what I have seen, previous studies on assessing the impacts of climate change have mostly used lower discount factors. Also, to my knowledge, results can be very sensitive to the choice of the discount factor. Could the authors justify the use of 5?

AUTHORS: *Indeed discounting is a critical aspect of the methodology and in our case we estimated the discount rates according to the EC Guide to Cost-Benefit Analysis of Investment Projects¹⁶ which we considered as the most reliable source. The discounting methodology is described in detail in the 'Cost-benefit analysis' of the Methods, but the key justification is given in these lines:*

'Here the Ramsey equation is calibrated using average growth rates for consumption per capita. As suggested in the EC Guide to Cost-Benefit Analysis of Investment Projects¹⁹, we distinguish

between the so-called Cohesion countries, which benefit from the Cohesion Fund, and the rest of the EU Member States²⁰. For the 2014-2020 period, the Cohesion Fund concerns Bulgaria, Croatia, Cyprus, the Czech Republic, Estonia, Greece, Hungary, Latvia, Lithuania, Malta, Poland, Portugal, Romania, Slovakia and Slovenia. From our macroeconomic projections the average growth rate of consumption per capita for cohesion countries is equal to 2%, while for the rest of the countries is 1% per year.

We further assume a value of 1 for ρ , which is chosen as a central value between 0, i.e. no preference for current or future generations, and 2, which is the value attributed by Weitzman in his review of the Stern Review²¹. Values of η in literature typically range between 1 and 4, with a central estimate of 2²², which is the value that we assume here.

With these values for g , ρ and η , the resulting discount rates are 5% for the cohesion countries (poorer countries in Europe) and 3% for the other Member States. The resulting discount rates appear to be in line with those suggested by the European Commission for the cost-benefit analysis of major investment projects^{21,22}.

In addition we have repeated the analysis without discounting in order to discuss the impact of discounting on the results (see last paragraph of the manuscript):

‘The use of discounting is another critical aspect as high discount rates put more weight on short term costs and benefits. Here this means that capital investment costs in protection, which occur now, are emphasized, while future benefits of adaptation are downgraded. In sum this may discourage taking action now. For the same analysis without discounting the estimated dyke height is on average 4-11 cm higher along the European coastline, compared to when discounting costs and benefits. This would require an additional mean annual cost of 1.41 billion € and 2.21 billion € during this century, depending on the scenario. The benefits of the increased protection would be that an additional 7.5% to 9.7% of Europe’s coastline would be protected to rising seas, and the EAD and EAPF by the end of the century would be further reduced by 5.1-16.4 billion € and 40 to 255 thousand people, respectively, compared to when not discounting.’

- From this type of work I would certainly expect some discussion on the implications of those results in terms of policy. I believe such a discussion (even if it only were a couple of lines long) would certainly strengthen the manuscript.

AUTHORS: The work was done in collaboration of DG CLIMA of the European Commission and therefore was designed in support of their climate policy. The most relevant findings to EU policymakers are that GHG emissions mitigation can limit coastal flood losses substantially, and the climate change adaptation can further considerably reduce coastal flood risk in a cost-efficient way. The above have been added in the last paragraphs of the paper. We are confident that together with the information about the results without discount rates and the fact that the methodology can be tailored to different levels of risk perception, the manuscript is very informative for policy makers.

‘Our results highlight potential savings in terms of avoided damages that can be obtained through strategies that increase structural flood protection at the subnational scale. Large-scale studies as the one presented help to initiate the dialogue with stakeholders and identify priority regions. For the actual implementation of individual measures at local scale, they should be complemented by detailed studies that are performed using local models and data. Local studies are also more appropriate to evaluate the co-existence of hard protection with other adaptation practices, such as nature-based solutions, retreat or accommodate. This requires very detailed planning using local data not publicly available at European scale. By no means we suggest that hard protection is the only or preferable option, and our analysis does not exclude the parallel implementation of more

sustainable environmental practices to enforce the physical and ecological resilience of coastal zones⁵. In addition we are by no means we are considering hard protection as the only or preferable adaptation option. It has been the most common approach until now and for that reason we consider the present analysis useful for practitioners and policy makers.

The applied probabilistic framework allows decision makers to interpret the results according to the amount of risk they consider as acceptable. Our projections of future coastal hazard and risk, as well as dyke costs, come with uncertainty, and in this report we evaluated the adaptation option that optimizes the expected benefits vs costs. However, stakeholders could select a more conservative criterion and optimize adaptation investments in view of high-end, less probable future scenarios, under which flood impacts will be higher, instead of the whole range currently considered. Such a choice would result in higher adaptation costs, but would also imply lower risks for future generations, as the analysis would prioritize protection against the rarer and more catastrophic events.

The use of discounting is another critical aspect as high discount rates put more weight on short term costs and benefits. Here this means that capital investment costs in protection, which occur now, are emphasized, while future benefits of adaptation are downgraded. In sum this may discourage taking action now. For the same analysis without discounting the estimated dyke height is on average 4-11 cm higher along the European coastline, compared to when discounting costs and benefits. This would require an additional mean annual cost of 1.41 billion € and 2.21 billion € during this century, depending on the scenario. The benefits of the increased protection would be that an additional 7.5% to 9.7% of Europe's coastline would be protected to rising seas, and the EAD and EAPF by the end of the century would be further reduced by 5.1-16.4 billion € and 40 to 255 thousand people, respectively, compared to when not discounting.'

- The manuscript leaves the impression that hard protection is a clear no-brain solution to rising ESLs because it is generally economically beneficial. However, as the authors correctly note in the introduction, hard protection often results in other problems. It is also not always desirable (e.g. in tourist areas, for example in the Mediterranean). Further, decisions are not only made on monetary considerations but, for example, social costs are also considered. The discussion on these points is very limited in the manuscript and may communicate the wrong message to the readers.

AUTHORS: The comment is in line with the view of reviewer 2 and for that reason in the revision we have expanded the discussion of all adaptation pathways and we highlight that we are not excluding them as viable options. This has been clarified in the introduction:

'There exists a range of possible adaptation measures to increase the resilience of future coastal societies to flooding²³, summarized as protect, accommodate, retreat and do nothing²⁴. The latter option seems unrealistic considering the substantial presence of critical infrastructure along the global coastline^{25,26}, and the projected economic losses from coastal flooding^{7,14}. Retreat involves relocating dwellings and infrastructure in order to reduce coastal flood risk²⁷, but relocation is often challenging due to public opposition or practical limitations (e.g. moving critical infrastructure such as ports of power plants)¹²⁸. Accommodate involves reducing the damages from coastal flooding, either through effective forecasting/warning systems and emergency response²⁹, or flood proofing of structures²⁴. Protect involves preventing flooding through natural (dunes) or artificial (dykes) structures, beach nourishment, and nature-based solutions, which have recently gained attention as more environmentally sustainable ways to protect and maintain coastlines³⁰.

Among the various adaptation options, hard protection is the strategy that delivers the most predictable levels of safety against coastal extremes and sea level rise and has been applied widely along developed coastlines of Europe⁵, despite the fact that hard protection can affect the landscape

in a negative way, increase erosion, reduce amenity value and result in more catastrophic events in the case of failure³¹. As a result, there is the increasing trend to apply a combination of adaptation options, with some kind of impermeable protection element being one of them.

Here, we evaluate the costs and benefits of applying additional protection through dyke improvements along the European coastline, assuming that the densely populated and high income European coastal communities will choose to 'hold the line'.

And the discussion:

'Our results highlight potential savings in terms of avoided damages that can be obtained through strategies that increase structural flood protection at the subnational scale. Large-scale studies as the one presented help to initiate the dialogue with stakeholders and identify priority regions. For the actual implementation of individual measures at local scale, they should be complemented by detailed studies that are performed using local models and data. Local studies are also more appropriate to evaluate the co-existence of hard protection with other adaptation practices, such as nature-based solutions, retreat or accommodate. This requires very detailed planning using local data not publicly available at European scale. By no means we suggest that hard protection is the only or preferable option, and our analysis does not exclude the parallel implementation of more sustainable environmental practices to enforce the physical and ecological resilience of coastal zones⁵. In addition we are by no means we are considering hard protection as the only or preferable adaptation option. It has been the most common approach until now and for that reason we consider the present analysis useful for practitioners and policy makers.'

- I was a bit surprised to see rather high CB ratios in poorer, not very densely populated coasts (e.g south Italy). I imagine that these may result from e.g. large cities that are located in those administrative units? No need to address this in the manuscript, just a question from my side.

AUTHORS: Italy has some densely populated coastal areas in the south that can dominate the NUTS2 costs and benefits. Another important factor is that SLR in the Mediterranean is comparable to (or even higher than) the astronomical and meteorological tides. Instead in macro-tidal and more energetic coastlines the projected SLR is definitely lower than the expected astronomical and meteorological tides. The result is that in microtidal areas SLR will result in a strong increase in the frequency of flooding, and thus a large rise in damage and consequent benefits in case of further protection.

- Just a suggestion: I would not use the word "incentive" in the title as "incentive" usually implies receiving something in return. I would opt for "benefits" or "motivation". But this is up to the authors and editor to decide

AUTHORS: We have replaced 'incentive' with 'motivation' as suggested by the reviewer.

Finally, two typos that I noticed:

- line 60: i find the wording in point (v) confusing, it reads as if the authors also calculate river flood damages. Maybe replace "follow" with "following"?

AUTHORS: The text has been corrected accordingly

- Table 2, legend: "people flooded ... to coastal flooding" - should probably be "exposed"?

AUTHORS: We had several internal discussions about this and we decided to avoid the word 'exposed', as exposure has a different meaning in terms of IPCC.

I hope that the authors find my comments useful for improving the manuscript.

AUTHORS: Definitely and they are greatly appreciated.

Reviewer 2

This manuscript is good piece of work and it can be suitable for publication after the solution of two specific aspects:

1) I found the same manuscript (from the same authors) online published in the following web page:

<https://eartharxiv.org/wu4r6/>

No idea about how this works (is this possible?), but my major concern is a possible duplicity. Why duplicity? : that manuscript or preprint has a doi: 10.31223/osf.io/wu4r6

If Nature communications accept this manuscript, then I think we have the same manuscript published two times.

AUTHORS: The manuscript the reviewer refers to is the submitted version of the paper that was uploaded to a pre-print server. This is actually encouraged by Nature Research journals (see link below). The selected pre-print server is further among the recommended ones and we have provided the link of the preprint to the editor on submission). More details can be found in this link: <https://www.nature.com/nature-research/editorial-policies/preprints-and-conference-proceedings>

2) Along the first part of the manuscript were presented some "adaptation" measures. These measures are not really adaptation measures. Some of them are protection measures. I suggest a deep update about the existing management strategies, here some help.

Protection

This approach consists in the preservation of vulnerable areas (i.e., population centers, economic activities and resources) by means of constructing hard structures, or alternatively soft protection measures which does include beach restoration. Both hard and soft approaches seek to eliminate or mitigate damage to the coast by use of barriers to reduce wave impact and control hydraulic processes that influence sediment transport.

Accommodation

Accommodation includes all approaches for addressing flooding by revising and reorganizing human activities in the coastal zone, or even at the river basin level. This approach allows acceptance of a certain degree of flooding and erosion by construction methods, changing land use, and improving preparedness. The above means a continued usage of land at risk without attempting to prevent the area from being damaged by natural events, and allowing conservation and migration of ecosystems.

Use of ecosystems

This approach is based on the natural environmental influences over processes related to flooding (e.g., sediment capture and energy attenuation) by restoring or creating ecosystems. The use of ecosystems for flooding management is an integrated process to conserve and improve ecosystem health that sustains ecosystem services for human well-being.

Managed/planned retreat

Managed/planned retreat has the primary objective of reducing population, property, and infrastructure at risk through the planned withdrawal from coastal hazard zones, including flood-prone areas. Specific to the flooding hazard, managed retreat implements mitigation tools designed to move existing and proposed development out of the path of both short- and long-term coastal flooding. The approach may include managed realignment where shore-protection structures are removed selectively to allow natural coastal environments to be re-established.

Regards,

AUTHORS: We would like to thank the reviewer for the comment and the information. It is not our intention to present protection as the only solution or to neglect its potential negative implications. However, to date protection has been the most commonly used risk reduction (adaptation) strategy along Europe's coastlines. It is therefore also the most studied and documented (e.g., in terms of costs estimates) strategy. This allowed us in this first pan-European study on coastal adaptation to assess costs and benefits of coastal protection. We agree with the reviewer that other adaptation strategies, such as retreat or accommodate, could play an important role, but assessing them at European scale is challenging given the paucity of information on the costs of implementation and the need for very local data required for the planning of such measures.

Relocation of homes and people often involves delicate decisions, while in the case of critical infrastructure it is often practically impossible (e.g. nuclear plants, sea ports). Accommodate does not avoid floods to happen but can considerably reduce damage to buildings and infrastructure. Not all building can be flood-proofed, while the costs can vary strongly among assets. Assessing accommodate requires high detail information of the nearshore assets, while cost estimates and risk reduction potential vary strongly. Recently, nature-based solutions (NBS) have gained more attention, yet to date there are few well documented applications along developed coastlines. In addition NBS is not a unique approach, but rather it comprises a wide range of measures, of which the effectiveness of their implementation depends on local characteristics that are difficult to integrate in a pan-European study.

For the above reasons we focus on hard protection, but we now discuss more extensively also other adaptation pathways given the word limit of the manuscript. We have expanded relevant paragraphs of the introduction as well as the discussion:

New text in the introduction:

‘There exists a range of possible adaptation measures to increase the resilience of future coastal societies to flooding²³, summarized as protect, accommodate, retreat and do nothing²⁴. The latter option seems unrealistic considering the substantial presence of critical infrastructure along the global coastline^{25,26}, and the projected economic losses from coastal flooding^{7,14}. Retreat involves relocating dwellings and infrastructure in order to reduce coastal flood risk²⁷, but relocation is often challenging due to public opposition or practical limitations (e.g. moving critical infrastructure such as ports of power plants)¹²⁸. Accommodate involves reducing the damages from coastal flooding, either through effective forecasting/warning systems and emergency response²⁹, or flood proofing of structures²⁴. Protect involves preventing flooding through natural (dunes) or artificial (dykes) structures, beach nourishment, and nature-based solutions, which have recently gained attention as more environmentally sustainable ways to protect and maintain coastlines³⁰.

Among the various adaptation options, hard protection is the strategy that delivers the most predictable levels of safety against coastal extremes and sea level rise and has been applied widely along developed coastlines of Europe⁵, despite the fact that hard protection can affect the landscape

in a negative way, increase erosion, reduce amenity value and result in more catastrophic events in the case of failure³¹. As a result, there is the increasing trend to apply a combination of adaptation options, with some kind of impermeable protection element being one of them.

Here, we evaluate the costs and benefits of applying additional protection through dyke improvements along the European coastline, assuming that the densely populated and high income European coastal communities will choose to 'hold the line'.

And the discussion:

'Our results highlight potential savings in terms of avoided damages that can be obtained through strategies that increase structural flood protection at the subnational scale. Large-scale studies as the one presented help to initiate the dialogue with stakeholders and identify priority regions. For the actual implementation of individual measures at local scale, they should be complemented by detailed studies that are performed using local models and data. Local studies are also more appropriate to evaluate the co-existence of hard protection with other adaptation practices, such as nature-based solutions, retreat or accommodate. This requires very detailed planning using local data not publicly available at European scale. By no means we suggest that hard protection is the only or preferable option, and our analysis does not exclude the parallel implementation of more sustainable environmental practices to enforce the physical and ecological resilience of coastal zones⁵. In addition we are by no means we are considering hard protection as the only or preferable adaptation option. It has been the most common approach until now and for that reason we consider the present analysis useful for practitioners and policy makers.'

References

- 1 Boettle, M., Rybski, D. & Kropp, J. P. Quantifying the effect of sea level rise and flood defence – a point process perspective on coastal flood damage. *Nat. Hazards Earth Syst. Sci.* **16**, 559-576, doi:10.5194/nhess-16-559-2016 (2016).
- 2 Vousdoukas, M. I. *et al.* Climatic and socioeconomic controls of future coastal flood risk in Europe. *Nature Climate Change*, doi:10.1038/s41558-018-0260-4 (2018).
- 3 Prah, B. F., Boettle, M., Costa, L., Kropp, J. P. & Rybski, D. Damage and protection cost curves for coastal floods within the 600 largest European cities. *Scientific Data* **5**, 180034, doi:10.1038/sdata.2018.34 (2018).
- 4 Batista e Silva, F., Lavalle, C. & Koomen, E. A procedure to obtain a refined European land use/cover map. *J. Land Use Sci.* **8**, 255-283, doi:10.1080/1747423X.2012.667450 (2012).
- 5 Jones, B. & O'Neill, B. C. Spatially explicit global population scenarios consistent with the Shared Socioeconomic Pathways. *Environmental Research Letters* **11**, doi:10.1088/1748-9326/11/8/084003 (2016).
- 6 O'Neill, B. C. *et al.* A new scenario framework for climate change research: the concept of shared socioeconomic pathways. *Clim. Change* **122**, 387-400, doi:10.1007/s10584-013-0905-2 (2014).
- 7 van Vuuren, D. P. *et al.* A new scenario framework for Climate Change Research: scenario matrix architecture. *Clim. Change* **122**, 373-386, doi:10.1007/s10584-013-0906-1 (2014).
- 8 Tamura, M., Kumano, N., Yotsukuri, M. & Yokoki, H. Global assessment of the effectiveness of adaptation in coastal areas based on RCP/SSP scenarios. *Clim. Change* **152**, 363-377, doi:10.1007/s10584-018-2356-2 (2019).
- 9 Lincke, D. & Hinkel, J. Economically robust protection against 21st century sea-level rise. *Global Environmental Change* **51**, 67-73, doi:https://doi.org/10.1016/j.gloenvcha.2018.05.003 (2018).
- 10 Hinkel, J. *et al.* Coastal flood damage and adaptation costs under 21st century sea-level rise. *Proceedings of the National Academy of Sciences* **111**, 3292-3297, doi:10.1073/pnas.1222469111 (2014).
- 11 Paprotny, D., Morales-Nápoles, O., Vousdoukas, M. I., Jonkman, S. N. & Nikulin, G. Accuracy of pan-European coastal flood mapping. *Journal of Flood Risk Management in press*, doi:doi:10.1111/jfr3.12459 (2018).
- 12 Howard, T., Lowe, J. & Horsburgh, K. Interpreting Century-Scale Changes in Southern North Sea Storm Surge Climate Derived from Coupled Model Simulations. *J. Clim.* **23**, 6234-6247, doi:10.1175/2010JCLI3520.1 (2010).
- 13 Weisse, R., von Storch, H., Niemeyer, H. D. & Knaack, H. Changing North Sea storm surge climate: An increasing hazard? *Ocean and Coastal Management* **68**, 58-68 (2012).
- 14 Vousdoukas, M. I. *et al.* Developments in large-scale coastal flood hazard mapping. *Natural Hazards and Earth System Science* **16**, 1841-1853, doi:10.5194/nhess-16-1841-2016 (2016).
- 15 Scussolini, P. *et al.* FLOPROS: an evolving global database of flood protection standards. *Nat. Hazards Earth Syst. Sci.* **16**, 1049-1061, doi:10.5194/nhess-16-1049-2016 (2016).
- 16 EC. *Guide to Cost-Benefit Analysis of Investment Projects - Economic appraisal tool for Cohesion Policy 2014-2020.* (2015).
- 17 Sartori, D. *et al.* *Guide to Cost-Benefit Analysis of Investment Projects. Economic appraisal tool for Cohesion Policy 2014-2020.* (European Commission, 2014).
- 18 Weitzman, M. L. A Review of the Stern Review on the Economics of Climate Change. *Journal of Economic Literature* **45**, 703-724, doi:doi: 10.1257/jel.45.3.703 (2007).
- 19 Gollier, C. & Hammitt, J. K. The Long-Run Discount Rate Controversy. *Annual Review of Resource Economics* **6**, 273-295, doi:10.1146/annurev-resource-100913-012516 (2014).

- 1 Hinkel, J., Nicholls, R., Vafeidis, A., Tol, R. J. & Avagianou, T. Assessing risk of and
adaptation to sea-level rise in the European Union: an application of DIVA. *Mitig Adapt
Strateg Glob Change* **15**, 703-719, doi:10.1007/s11027-010-9237-y (2010).
- 2 Ciscar, J. *et al.* *Climate Impacts in Europe. The JRC PESETA II Project.* (2014).
- 3 Meehl, G. A. *et al.* in *Climate Change 2007: The Physical Science Basis. Contribution of
Working Group I to the Fourth Assessment Report of the Intergovernmental Panel on
Climate Change* (eds S. Solomon *et al.*) 747–846 (Cambridge University Press, 2007).
- 4 Lincke, D. & Hinkel, J. Economically robust protection against 21st century sea-level rise.
Global Environmental Change **51**, 67-73,
doi:https://doi.org/10.1016/j.gloenvcha.2018.05.003 (2018).
- 5 Oppenheimer, M., Glavovic, B., Hinkel, J., Wal, R. van de, Magnan, A.K., Abd-Elgawad,
A., Cai, R., Cifuentes-Jara, M., Deconto, R.M., Ghosh, T., Hay, J., Isla, F., Marzeion, B.,
Meysignac, B., Sebesvari, Z. in *IPCC Special Report on the Ocean and Cryosphere in a
Changing Climate* (ed H.-O. *et al.* Pörtner) (Cambridge University Press, 2019).
- 6 Boettle, M., Rybski, D. & Kropp, J. P. Quantifying the effect of sea level rise and flood
defence – a point process perspective on coastal flood damage. *Nat. Hazards Earth Syst. Sci.*
16, 559-576, doi:10.5194/nhess-16-559-2016 (2016).
- 7 Vousdoukas, M. I. *et al.* Climatic and socioeconomic controls of future coastal flood risk in
Europe. *Nature Climate Change*, doi:10.1038/s41558-018-0260-4 (2018).
- 8 Prah, B. F., Boettle, M., Costa, L., Kropp, J. P. & Rybski, D. Damage and protection cost
curves for coastal floods within the 600 largest European cities. *Scientific Data* **5**, 180034,
doi:10.1038/sdata.2018.34 (2018).
- 9 Batista e Silva, F., Lavalle, C. & Koomen, E. A procedure to obtain a refined European land
use/cover map. *J. Land Use Sci.* **8**, 255-283, doi:10.1080/1747423X.2012.667450 (2012).
- 10 Jones, B. & O’Neill, B. C. Spatially explicit global population scenarios consistent with the
Shared Socioeconomic Pathways. *Environmental Research Letters* **11**, doi:10.1088/1748-
9326/11/8/084003 (2016).
- 11 O’Neill, B. C. *et al.* A new scenario framework for climate change research: the concept of
shared socioeconomic pathways. *Clim. Change* **122**, 387-400, doi:10.1007/s10584-013-
0905-2 (2014).
- 12 van Vuuren, D. P. *et al.* A new scenario framework for Climate Change Research: scenario
matrix architecture. *Clim. Change* **122**, 373-386, doi:10.1007/s10584-013-0906-1 (2014).
- 13 Tamura, M., Kumano, N., Yotsukuri, M. & Yokoki, H. Global assessment of the
effectiveness of adaptation in coastal areas based on RCP/SSP scenarios. *Clim. Change* **152**,
363-377, doi:10.1007/s10584-018-2356-2 (2019).
- 14 Hinkel, J. *et al.* Coastal flood damage and adaptation costs under 21st century sea-level rise.
Proceedings of the National Academy of Sciences **111**, 3292-3297,
doi:10.1073/pnas.1222469111 (2014).
- 15 Paprotny, D., Morales-Nápoles, O., Vousdoukas, M. I., Jonkman, S. N. & Nikulin, G.
Accuracy of pan-European coastal flood mapping. *Journal of Flood Risk Management* **in
press**, doi:doi:10.1111/jffr3.12459 (2018).
- 16 Howard, T., Lowe, J. & Horsburgh, K. Interpreting Century-Scale Changes in Southern
North Sea Storm Surge Climate Derived from Coupled Model Simulations. *J. Clim.* **23**,
6234-6247, doi:10.1175/2010JCLI3520.1 (2010).
- 17 Weisse, R., von Storch, H., Niemeier, H. D. & Knaack, H. Changing North Sea storm surge
climate: An increasing hazard? *Ocean and Coastal Management* **68**, 58-68 (2012).
- 18 Scussolini, P. *et al.* FLOPROS: an evolving global database of flood protection standards.
Nat. Hazards Earth Syst. Sci. **16**, 1049-1061, doi:10.5194/nhess-16-1049-2016 (2016).
- 19 EC. *Guide to Cost-Benefit Analysis of Investment Projects - Economic appraisal tool for
Cohesion Policy 2014-2020.* (2015).
- 20 Sartori, D. *et al.* *Guide to Cost-Benefit Analysis of Investment Projects. Economic appraisal
tool for Cohesion Policy 2014-2020.* (European Commission, 2014).

- 21 Weitzman, M. L. A Review of the 'Stern Review on the Economics of Climate Change'. *Journal of Economic Literature* **45**, 703-724, doi:doi: 10.1257/jel.45.3.703 (2007).
- 22 Gollier, C. & Hammitt, J. K. The Long-Run Discount Rate Controversy. *Annual Review of Resource Economics* **6**, 273-295, doi:10.1146/annurev-resource-100913-012516 (2014).
- 23 Aerts, J. C. J. H. *et al.* Evaluating Flood Resilience Strategies for Coastal Megacities. *Science* **344**, 473-475, doi:10.1126/science.1248222 (2014).
- 24 Brown, S. *et al.* Shifting perspectives on coastal impacts and adaptation. *Nature Clim. Change* **4**, 752-755, doi:10.1038/nclimate2344 (2014).
- 25 Koks, E. E. *et al.* A global multi-hazard risk analysis of road and railway infrastructure assets. *Nature Communications* **10**, 2677, doi:10.1038/s41467-019-10442-3 (2019).
- 26 Monioudi, I. N. *et al.* Climate change impacts on critical international transportation assets of Caribbean Small Island Developing States (SIDS): the case of Jamaica and Saint Lucia. *Reg Environ Change* **18**, 2211-2225, doi:10.1007/s10113-018-1360-4 (2018).
- 27 Hino, M., Field, C. B. & Mach, K. J. Managed retreat as a response to natural hazard risk. *Nature Climate Change* **7**, 364, doi:10.1038/nclimate3252
<https://www.nature.com/articles/nclimate3252#supplementary-information> (2017).
- 28 Gibbs, M. T. Why is coastal retreat so hard to implement? Understanding the political risk of coastal adaptation pathways. *Ocean Coast. Manag.* **130**, 107-114, doi:https://doi.org/10.1016/j.ocecoaman.2016.06.002 (2016).
- 29 Kreibich, H. *et al.* Adaptation to flood risk: Results of international paired flood event studies. *Earth's Future* **5**, 953-965, doi:10.1002/2017EF000606 (2017).
- 30 Temmerman, S. *et al.* Ecosystem-based coastal defence in the face of global change. *Nature* **504**, 79, doi:10.1038/nature12859 (2013).
- 31 Pilkey, O. H. & Dixon, K. L. *The Corps and the Shore*. (Island Press, 1996).

REVIEWERS' COMMENTS:

Reviewer #1 (Remarks to the Author):

The authors have invested substantial effort in addressing all the comments of my review. I believe that the manuscript has improved considerably with those additions and would recommend publication.

Reviewer #2 (Remarks to the Author):

I think you did the homework and this manuscript is ready to be accepted.

keep the good work!

Regards,